# Comparison of the Effect of Sugammadex and Pyridostigmine on Postoperative Catheter-Related Bladder Discomfort: A Retrospective Matched Cohort Analysis

**DOI:** 10.3390/medicina58050590

**Published:** 2022-04-26

**Authors:** Young-Suk Kwon, Jong-Ho Kim, Sung-Mi Hwang, Jae-Wang Choi, Sang-Soo Kang

**Affiliations:** 1Department of Anesthesiology and Pain Medicine, Chuncheon Sacred Heart Hospital, College of Medicine, Hallym University, Chuncheon 24253, Korea; gettys@hallym.or.kr (Y.-S.K.); poik99@hallym.or.kr (J.-H.K.); 2Department of Anesthesiology and Pain Medicine, Kangdong Sacred Heart Hospital, College of Medicine, Hallym University, Seoul 05355, Korea; hkingka86@naver.com (J.-W.C.); kssege@naver.com (S.-S.K.)

**Keywords:** catheter-related bladder discomfort, pyridostigmine, sugammadex

## Abstract

*Background and Objectives*: As the use of sugammadex for reversing neuromuscular blockade during general anesthesia increases, additional effects of sugammadex have been reported compared to cholinesterase inhibitors. Here, we compare the incidence of postoperative catheter-related bladder discomfort (CRBD) between sugammadex and pyridostigmine/glycopyrrolate treatments for reversing neuromuscular blockade. *Materials and Methods*: We retrospectively analyzed patients aged ≥ 18 years who underwent surgery under general anesthesia, received sugammadex or pyridostigmine with glycopyrrolate to reverse neuromuscular blockade, and had a urinary catheter in the post-anesthesia care unit between March 2019 and February 2021. After applying the exclusion criteria, 1179 patients were included in the final analysis. The incidence and severity of CRBD were collected from post-anesthesia recovery records. *Results*: The incidence was 13.7% in the sugammadex group (*n* = 211) and 24.7% in the pyridostigmine group (*n* = 968). Following propensity score matching, 211 patients each were included in the pyridostigmine and sugammadex matched group (absolute standardized difference (ASD), 0.01–0.05). Compared to the pyridostigmine group, the odds ratio for CRBD occurring in the sugammadex group was 0.568 (95% confidential interval, 0.316–1.021, *p* = 0.059). *Conclusions*: Sugammadex has a similar effect on the occurrence of postoperative CRBD compared with pyridostigmine.

## 1. Introduction

As the use of sugammadex (a modified gamma-cyclodextrin) increases for the reversal of neuromuscular blockade during general anesthesia, its additional effects have been compared to those of cholinesterase inhibitors (e.g., pyridostigmine and neostigmine) [1,2,3]. Although cholinesterase inhibitors are used to activate nicotinic receptors, they also activate muscarinic receptors, so muscarinic antagonists are commonly used to manage the effects of excess acetylcholine [4].

Urinary catheterization for surgery is frequently essential; however, it can lead to catheter-related bladder discomfort (CRBD) in the postoperative period. This is a frequent postoperative patient complaint in the post-anesthesia care unit (PACU). The symptoms of CRBD include an urge to void or discomfort in the suprapubic area, which is caused by the stimulation of muscarinic receptors (particularly subtype M3) located in the bladder wall near the catheter [5,6]. Muscarinic antagonists help with symptom relief and prevention [7,8,9,10].

Unlike cholinesterase inhibitors, sugammadex selectively binds to steroidal neuromuscular blockers and inactivates them to reverse neuromuscular blockade. The muscarinic effect of sugammadex is limited; concomitant use of muscarinic antagonists is not required.

In this propensity score matching study, we compared the incidence of CRBD in the PACU between sugammadex and pyridostigmine/glycopyrrolate treatments for reversing neuromuscular blockade.

## 2. Materials and Methods

### 2.1. Patients

The study protocol was approved by the Clinical Research Ethics Committee of Chuncheon Sacred Heart Hospital, Hallym University (IRB No. CHUNCHEON 2021-06-024). All study data were acquired from the clinical data repository of Hallym University Medical Center. All patients were at least 18 years of age, had undergone surgery under only general anesthesia, received sugammadex or pyridostigmine with glycopyrrolate to reverse neuromuscular blockade, and had a urinary catheter while in the PACU between March 2019 and February 2021. General anesthesia was induced with propofol or etomidate with a neuromuscular blocking agent (rocuronium) and maintained by sevoflurane or desflurane or propofol with or without remifentanil. The patients were divided into a group given sugammadex (sugammadex group) and a group given pyridostigmine with glycopyrrolate (pyridostigmine group) to reverse neuromuscular blockade. Exclusion criteria were no use of sugammadex or pyridostigmine, combined use of sugammadex and pyridostigmine, use of glycopyrrolate (before and during anesthesia) in the sugammadex group, use of dexmedetomidine or ketamine during anesthesia [11], combined general and regional anesthesia, and insufficient patient data in the medical records

### 2.2. Variables

Matching was performed with consideration of age, sex, history of benign prostate hyperplasia and Parkinson’s disease, body mass index (kg/m^2^), timing of urinary catheter insertion (before or after anesthesia induction), size of urinary catheter, type of anesthetic agent (inhalation or total intravenous anesthesia), anesthesia time, atropine use during anesthesia, American Society of Anesthesiologists physical status, urological surgery status (endoscope used or not used), use of patient-controlled analgesia (PCA), and laparoscopic surgery status.

The severity of CRBD was classified as follows: “none”, “mild” (reported by patients only when asked), “moderate” (reported unprompted to a physician or nurse without any behavioral response), and “severe” (reported unprompted along with behavioral responses such as flailing limbs, strong vocalization, or attempts to pull out the catheter [12,13]. However, for the purposes of matching, only the incidence of CRBD was considered (none vs. mild–severe). The CRBD score was collected from post-anesthesia recovery records. The score was checked at admission and after 15 min by the nurse in charge. We chose the higher score of the two.

### 2.3. Statistics

Continuous data are presented as means and standard deviations. Categorical data are presented as frequencies and percentages. Odds ratios (unadjusted (UA), selectively adjusted for variables after backward elimination (SA; sex, catheter insertion timing, and catheter size), and fully adjusted for all variables and propensity scores (FA)), and 95% confidence intervals for the postoperative occurrence of CRBD were determined using logistic regression analysis. Propensity scores were calculated on the basis of pyridostigmine and sugammadex use. Selection bias was reduced via propensity score matching. Absolute standardized differences (ASD) were used to determine the influence of covariates on the two groups. Groups are generally considered similar when the standardized difference is <20% [14]. The rationale for using, and methods for applying, propensity scores when analyzing the exposure variable have been described elsewhere [15,16]. All *p*-values were two-sided, and *p*-values < 0.05 were considered statistically significant. IBM SPSS Statistics (version 26.0; IBM, Armonk, NY, USA) was used for the statistical analyses.

## 3. Results

### 3.1. Patient Characteristics

In total, 8345 patients received general anesthesia from March 2019 to February 2021. Among them, 2934 patients had a urinary catheter during anesthesia; 1693 patients aged ≥ 18 years, with a urinary catheter in the PACU and a CRBD score, were initially screened. After applying the exclusion criteria, 1179 patients were included in the final analysis.

### 3.2. Incidence of CRBD

Table 1 lists the incidence and severity of CRBD in patients who received sugammadex (*n* = 211) and those who received pyridostigmine (*n* = 968). The incidences were 13.7% and 24.7%, respectively. Table 2 presents the demographic characteristics and clinical variables of patients before and after matching. The ASD before matching was 0.02–0.79 across all variables, and that after matching was ≤0.05.

### 3.3. Odds Ratios for CRBD Occurrence in the Sugammadex Group

Table 3 lists odds ratios before propensity matching and Table 4 lists odds ratios after matching. After matching, compared to the pyridostigmine group, the UA, FA, and SA odds ratios for CRBD occurrence in the sugammadex group were 0.605 (*p* = 0.055), 0.568 (*p* = 0.059), and 0.544 (*p* = 0.039), respectively (Table 4).

## 4. Discussion

Urinary catheterization during various surgeries frequently leads to CRBD in the immediate postoperative period. The incidence of CRBD in the PACU ranges from 47–90% depending on the type of surgery [17]. CRBD is characterized by a burning sensation that spreads from the suprapubic area, which is associated with discomfort or the urge to void [12,17]. The cause of CRBD is catheter-induced bladder irritation related to muscarinic receptor (particularly subtype M3)-mediated involuntary contractions of the bladder and lower urinary tract (e.g., overactive bladder syndrome) [18,19]. Male sex, use of a urinary catheter size > 18 Fr, and urological surgery are independent predictors of CRBD [17,20]. Muscarinic antagonists have been studied for the prevention or treatment of CRBD; notable side effects include nausea, vomiting, sedation, and dry mouth [7,8,9,10,13]. In this study, we compared the effects of sugammadex and pyridostigmine on postoperative CRBD. These agents are essential for reversing neuromuscular blockade after general anesthesia.

Pyridostigmine is a cholinesterase inhibitor that also stimulates muscarinic receptors. Accordingly, glycopyrrolate and atropine are used as adjuncts. They have dissociable inhibitory effects on muscarinic receptors. Glycopyrrolate has high affinity at the M3 receptor, whereas atropine has less selectivity for the muscarinic receptor [10,21]. In a study of cholinesterase inhibitor adjuvants for the reversal of neuromuscular blockade, the incidence of CRBD was significantly lower in a glycopyrrolate than atropine group [9]. Glycopyrrolate is also effective for preventing CRBD when administered preoperatively as a premedication [8].

Sugammadex selectively binds to steroidal neuromuscular blockers and concomitant use of a muscarinic antagonist is not required. In our hospital, we use pyridostigmine with glycopyrrolate to reverse neuromuscular blockade. In this study, cases in which glycopyrrolate was used as a premedication or for other reasons during anesthesia were excluded from the sugammadex group. In the pyridostigmine group, glycopyrrolate was used in combination to reverse neuromuscular blockade, so all patients who used it for other reasons were included. The occurrence of CRBD in the pyridostigmine group presumably resulted from the muscarinic effect of pyridostigmine, combined with the competing antimuscarinic effect of glycopyrrolate. Note that other unwanted side effects of glycopyrrolate may occur. In this regard, it is meaningful that sugammadex, which does not require the concurrent use of anticholinergic drugs, had similar effects on CRBD.

Among studies comparing the effects of sugammadex and cholinesterase inhibitors, some considered the effects of glycopyrrolate, while others did not [1,2,3,22,23,24]. Because we have only been using pyridostigmine, we only compared sugammadex with pyridostigmine in this study. However, comparison with neostigmine would enhance the clinical impact of our findings. Kim et al. [3] reported that the odds ratios of postoperative nausea and vomiting were lower in their sugammadex than neostigmine group, although they did not differ significantly between their sugammadex and pyridostigmine groups. An et al. [1] reported that the anticholinergic effects of glycopyrrolate on bowel movements may compensate for the cholinergic side effects of pyridostigmine.

In the present study, although the incidence of CRBD was lower in the sugammadex group, after propensity score matching the odds ratio of FA did not differ significantly with regard to CRBD occurrence between the two groups.

It is insufficient to conclude that sugammadex is superior to pyridostigmine based only on the odds ratio of SA. A large-scale randomized controlled study may be needed for a more accurate comparison. Overall, in patients at high risk of CRBD, it remains important to select a neuromuscular blockade reversal agent that has little effect on CRBD. Because the severity of CRBD is significantly correlated with postoperative pain score, reducing the incidence of CRBD is important to reduce postoperative pain and increase patient satisfaction [25].

This study had some limitations. First, the effects of sugammadex and pyridostigmine doses were not analyzed; doses varied depending on neuromuscular blockade status and patient body weight. Second, in the pyridostigmine group, we included all cases given glycopyrrolate as a premedication or for other reasons during anesthesia. Additional use other than concomitant use with pyridostigmine may have influenced the occurrence of CRBD in the pyridostigmine group [6]. Third, we did not analyze drugs used in PCA separately: we only compared the presence or absence of PCA, and drugs such as nonsteroidal anti-inflammatory drugs may affect CRBD. Fourth, a larger study population will be required to confirm our results after matching. We added the odds ratio before matching because the data are suitable for the one-in-ten rule [26,27]. Fifth, in general, the prospective studies dealing with CRBD exclude the patient with a history of urinary tract infection, bladder outflow obstruction, and neurogenic bladder, but all were included in our study. This may also have influenced the results.

## 5. Conclusions

This multivariate analysis based on propensity score matching revealed no significant differences between sugammadex and pyridostigmine administration (for reversing neuromuscular blockade) on the occurrence of postoperative CRBD. However, a larger prospective randomized controlled study is needed to confirm our findings.

## Figures and Tables

**Table 1 medicina-58-00590-t001:** Incidence and severity of CRBD.

	Pyridostigmine (*n* = 968)	Sugammadex (*n* = 211)
None	728 (75.2)	182 (86.2)
Mild	184 (19.0)	18 (8.5)
Moderate	47 (4.8)	8 (3.7)
severe	9 (0.9)	3 (1.4)

Date are presented as number (%).

**Table 2 medicina-58-00590-t002:** Baseline patient characteristics and clinical data before and after matching.

	Before Matching			After Matching		
	Pyridostigmine (*n* = 968)	Sugammadex (*n* = 211)	ASD	Pyridostigmine (*n* = 211)	Sugammadex (*n* = 211)	ASD
Age (years)	60 [15]	71 [14]	0.79	71 [11]	71 [14]	0.01
Male sex	468 (48.3)	99 (46.9)	0.02	100 (47.4)	99 (46.9)	0.01
BMI (kg/m^2^)	24.8 [3.7]	24.1 [4.1]	0.16	24.3 [3.6]	24.1 [4.1]	0.05
Anesthesia time (min)	128.2 [66.3]	147.3 [90.9]	0.21	146 [73.9]	147.3 [90.9]	0.01
BPH	121 (12.5)	38 (18)	0.14	35 (16.6)	38 (18)	0.03
Parkinson’s Ds	10 (1)	5 (2.4)	0.08	6 (2.8)	5 (2.4)	0.03
Catheter insertion timing			0.27			0.00
before	463 (47.8)	129 (61.1)		129 (61.1)	129 (61.1)	
after	505 (52.2)	82 (38.9)		82 (38.9)	82 (38.9)	
Catheter size (≥18 Fr)	68 (7)	13 (6.2)	0.03	12 (5.7)	13 (6.2)	0.02
Anesthesia agent			0.12			0.02
inhalation	880 (90.9)	198 (93.8)		199 (94.3)	198 (93.8)	
TIVA	88 (9.1)	13 (6.2)		12 (5.7)	13 (6.2)	
Atropine use	23 (2.4)	7 (3.3)	0.05	9 (4.3)	7 (3.3)	0.05
ASA PS 3-4	765 (79)	201 (95.3)	0.76	199 (94.3)	201 (95.3)	0.04
PCA	533 (55.1)	163 (77.3)	0.52	168 (79.6)	163 (77.3)	0.05
Urology surgery						
not used scope	208 (21.5)	30 (14.2)	0.2	30 (14.2)	30 (14.2)	0.00
used scope	158 (16.3)	23 (10.9)	0.17	23 (10.9)	23 (10.9)	0.00
Laparoscopic surgery	105 (10.8)	34 (16.1)	0.14	32 (15.2)	34 (16.1)	0.02

Catheter insertion timing before = before anesthesia induction, after = after anesthesia induction, TIVA = total intravenous anesthesia, ASA PS = American Society of Anesthesiologists physical status, PCA = patient-controlled analgesia, ASD = absolute standardized difference (expressed to two decimal places). Date are presented as number (%) or mean (SD).

**Table 3 medicina-58-00590-t003:** Odds of CRBD in the sugammadex group, before propensity score matching between the pyridostigmine and sugammadex group.

		Odds Ratio (95% CI)	*p*-Value
UA	Pyridostigmine	Reference	
	sugammadex	0.483 (0.318–0.734)	0.001
FAS	Pyridostigmine	Reference	
	sugammadex	0.646 (0.397–1.052)	0.079
SA	Pyridostigmine	Reference	
	sugammadex	0.611 (0.381–0.979)	0.041

UA = unadjusted, FAS = adjusted for all variables and propensity score, SA = selective adjusted.

**Table 4 medicina-58-00590-t004:** Odds of CRBD in the sugammadex group, following propensity score matching between the pyridostigmine and sugammadex group.

		Odds Ratio (95% CI)	*p*-Value
UA	Pyridostigmine	Reference	
	sugammadex	0.605 (0.362–1.011)	0.055
FAS	Pyridostigmine	Reference	
	sugammadex	0.568 (0.316–1.021)	0.059
SA	Pyridostigmine	Reference	
	sugammadex	0.544 (0.306–0.969)	0.039

UA = unadjusted, FAS = adjusted for all variables and propensity score, SA = selective adjusted.

## Data Availability

The datasets used in this current study are available from the corresponding author on reasonable request.

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
