# Peer review of "Comparison of the Effect of Sugammadex and Pyridostigmine on Postoperative Catheter-Related Bladder Discomfort: A Retrospective Matched Cohort Analysis"

_medicina, 2022, doi:10.3390/medicina58050590_

Round 1

Reviewer 1 Report

I read with great interest the article titled “Comparison of the effect of sugammadex and pyridostigmine on postoperative catheter-related bladder discomfort: A retrospective matched cohort analysis,” in which the authors concluded that sugammadex has similar effect on the occurrence of postoperative CRBD when compared with pyridostigmine.

Comments and suggestions:

  • Was the catheterization done in the PACU? (Lines 55- 56 states that “you included patients who had undergone surgery under general anesthesia, and underwent urinary catheterization in the PACU.” This differs from what is written in the results section. Please clarify this point, and if needed, please correct it here and in the abstract.
  • Since you included all surgeries, did you include all urologic surgeries, including those in which bladder related discomfort might be related to surgical stimuli and instrumentation?
  • You did not mention in the inclusion and exclusion criteria whether patients who underwent combined general anesthesia and neuraxial or regional anesthesia were included (For instance, using both epidural and general anesthesia).
  • Did you include patients who underwent multiple trials for catheter insertion? What type of catheter was inserted?
  • Did you exclude patients with history of urinary tract infection, bladder outflow obstruction, and neurogenic bladder?
  • In your results section, you compared TIVA and inhalational anesthesia as maintenance for anesthesia, yet you did not include the agents used. For example, was Ketamine used in the induction or maintenance of anesthesia in those patients?
    Please refer to the following article:
    Hu B, Li C, Pan M, Zhong M, Cao Y, Zhang N, Yuan H, Duan H. Strategies for the prevention of catheter-related bladder discomfort: A PRISMA-compliant systematic review and meta-analysis of randomized controlled trials. Medicine (Baltimore). 2016 Sep;95(37):e4859. doi: 10.1097/MD.0000000000004859.
  • A detailed description of the anesthetic methods and perioperative analgesia must be written in the methodology section. Also, the PACU care and documentation method must be described in a more detailed manner. Were the patients checked every 10 minutes for catheter-related bladder discomfort? How was that documented? Do you have a field on the PACU charts for documenting these incidents, or was the data retrieved mainly from the nursing written notes?
  • In line 58, you mentioned that you excluded those who administered glycopyrrolate before and during anesthesia. Why didn’t you exclude patients who administered atropine, too? Although glycopyrrolate has high affinity at the M3 receptor, whereas atropine has less selectivity for the muscarinic receptor, it still might affect the rate of postoperative catheter-related bladder discomfort.
  • It is mentioned in the discussion (lines 138-142) that “In this study, cases involving atropine use were excluded from both groups; cases involving glycopyrrolate use as premedication or during anesthesia were excluded from the sugammadex group. In the pyridostigmine group, all cases involving glycopyrrolate use as premedication or during surgery were included, in addition to cases involving concomitant administration with pyridostigmine.” This needs further detailed clarification in the methods and results section as it differs from what has been previously mentioned.

Author Response

Thanks for your works and comment. 

Please see the attached file..

Reviewer 2 Report

I read the manuscript carefully.

Postoperative catheter-related bladder discomfort (CRBD) is a serious complication of surgery, especially in men.
Many studies exist on the causes of CRBD, its frequency of occurrence with different anesthesia methods, and prevention and treatment strategies.
The current study is about CRBD between the differences in drugs used for muscle relaxant reversal. 
This is an interesting study on the subject.
On the other hand, the reviewers believe that there are several points that the authors should approach to improve this manuscript.
The following is a list.

Conclusion of this study
The authors provided a result that "Compared to the pyridostigmine group, the odds ratios for CRBD occurrence in the sugammadex group were 0.568 (95% confidential interval;  0.316-1.021, P = 0.059)"
Dependent on the result, they concluded that " Conclusions: Our findings suggest that sugammadex has similar effect on the occurrence of postoperative CRBD when compared with pyridostigmine".

On the other hand, as shown in Table 3, the Odds ratio is 0.544 (P-value 0.039) under SA conditions.

Considering these factors, the sugammadex group can be superior to the pyridostigmine group.
Please discuss this issue clearly.

Calculation of the number of patients
1,179 patients were finally included in the analysis of this study.
Discuss whether this number of patients is sufficient to draw conclusions

Postoperative analgesia
The only description of postoperative analgesia is that of PCA.
If the authors have results on the use of NSIAD, please include them in your study.
NSAID has been shown to reduce CRBD.

Author Response

Thanks for your works and comment. 

Please see the attached file...... 

Round 2

Reviewer 1 Report

I would like to thank the authors for their responses and their efforts in improving their manuscript. All comments were addressed by them in their responses file.